# Similarity-based cooperation

## Abstract

As machine learning agents act more autonomously in the world, they will increasingly interact with each other. Unfortunately, in many social dilemmas like the one-shot Prisoner's Dilemma, standard game theory predicts that ML agents will fail to cooperate with each other. Prior work has shown that one way to enable cooperative outcomes in the one-shot Prisoner's Dilemma is to make the agents mutually transparent to each other, i.e., to allow them to access one another's source code (Rubinstein, 1998; Tennenholtz, 2004) – or weights in the case of ML agents. However, full transparency is often unrealistic, whereas partial transparency is commonplace. Moreover, it is challenging for agents to learn their way to cooperation in the full transparency setting. In this paper, we introduce a more realistic setting in which agents only observe a single number indicating how similar they are to each other. We prove that this allows for the same set of cooperative outcomes as the full transparency setting. We also demonstrate experimentally that cooperation can be learned using simple ML methods.

## 1 Introduction

As AI systems start to autonomously interact with the world, they will also increasingly interact with each other. We already see this in contexts such as trading agents (CFTC & SEC, 2010), but the number of domains where separate AI agents interact with each other in the world is sure to grow; for example, consider autonomous vehicles. In the language of game theory, AI systems will play general-sum games with each other. For example, autonomous vehicles may find themselves in Game-of-Chicken-like dynamics with each other (cf. Fox et al., 2018). In many of these interactions, cooperative or even peaceful outcomes are not a given. For example, standard game theory famously predicts and recommends defecting in the one-shot Prisoner's Dilemma. Even when cooperative equilibria exist, there are typically many equilibria, including uncooperative and asymmetric ones. For instance, in the infinitely repeated Prisoner's Dilemma, mutual cooperation is played in some equilibria, but so is mutual defection, and so is the strategy profile in which one player cooperates 70% of the time while the other cooperates 100% of the time. Moreover, the strategies from different equilibria typically do not cooperate with each other. A recent line of work at the intersection of AI/(multi-agent) ML and game theory aims to increase AI/ML systems' ability to cooperate with each other (Stastny et al., 2021; Dafoe et al., 2021; Conitzer & Oesterheld, 2022).

Prior work has proposed to make AI agents *mutually transparent* to allow for cooperation in equilibrium (McAfee 1984; Howard 1988; Rubinstein 1998, Section 10.4; Tennenholtz 2004; Barasz et al. 2014; Critch 2019; Oesterheld 2019b). Roughly, this literature considers for any given 2-player normal-form game $\Gamma$ the following *program meta game*: Both players submit a computer program, e.g., some neural net, to choose actions in $\Gamma$ on their behalf. The computer program then receives as input the computer program submitted by the other player. Prior work has shown that the program meta game has cooperative equilibria in the Prisoner's Dilemma.

Unfortunately, there are multiple obstacles to cooperation based on full mutual transparency. 1) While partial transparency is the norm, settings of full transparency are rare. For example, while GPT-3's architecture and training regime are public knowledge, the exact learned model is not. 2) Games played with full transparency in general have many equilibria, including ones that are much worse for some or all players than the Nash equilibria of the underlying game (see the folk theorems given by Rubinstein 1998, Section 10.4, and Tennenholtz 2004). In particular, full mutual transparency can make the problem of equilibrium selection harder. 3) The full transparency setting poses challenges to modern ML methods. In particular, it requires at least one of the models to receive as input a model that has at least as many parameters as itself. Meanwhile, most modern

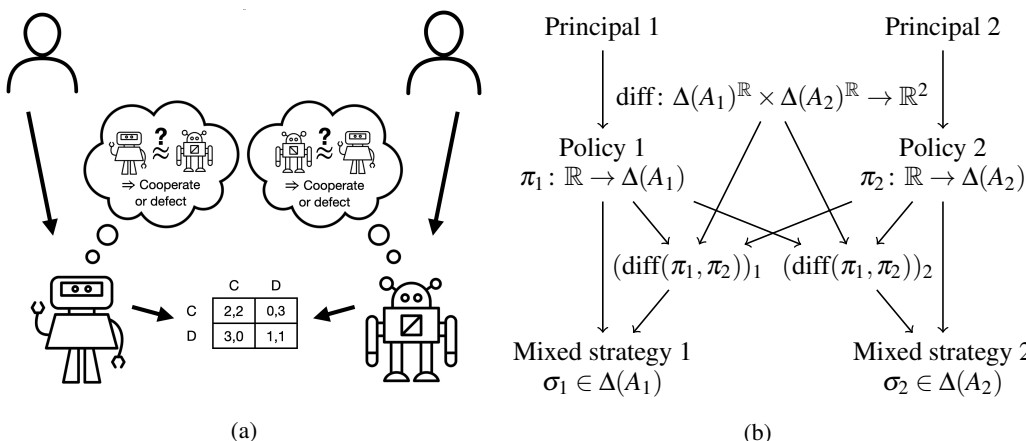

(a)             (b)

Figure 1: (a) Illustration of the diff meta game of a Prisoner's Dilemma. (b) A graphical representation of diff meta games (Definition 1). Nodes with two incoming nodes are determined by applying one of the parent nodes to the other.

successes of ML use models that are orders of magnitudes larger than the input. Consequently, we are not aware of successful projects on learning general-purpose models such as neural nets in the full transparency setting.

**Contributions** In this paper we introduce a novel variant of program meta games called *difference (diff) meta games* that enables cooperation in equilibrium while also addressing obstacles 1–3. As in the program meta game, we imagine that two players each submit a program or *policy* to instruct an *agent* to play a given game, such as the Prisoner's Dilemma. The main idea is that before choosing an action, the agents are given information about how *similar* the two players' policies are to each w.r.t. how they make the present decision. We formally introduce this setup in Section 3. For an informal illustration, see Figure 1a. Because it requires a much lower degree of mutual transparency, we find the diff meta game setup more realistic than the full mutual transparency setting. Thus, it addresses Obstacle 1 to cooperation based on full mutual transparency.

Diff meta games can still have cooperative equilibria when the underlying base game does not. Specifically, in Prisoner's Dilemma-like games, there are equilibria in which both players submit policies that cooperate with similar policies and thus with each other. We call this phenomenon *similarity-based cooperation (SBC)*. For example, consider the Prisoner's Dilemma as given in Table 1 for $G = 3$. (We study such examples in more detail in Section 3.) Imagine that the players can only submit *threshold policies* that cooperate if and only if the perceived difference to the opponent is at most $\theta_i$. As a measure of difference, the policies observe $\mathrm{diff}(\theta_1, \theta_2) = |\theta_1 - \theta_2| + N$, where $N$ is sampled independently for each player according to the uniform distribution over $[0, 1]$. For instance, if Player 1 submits a threshold of $1/2$ and Player 2 submits a threshold of $3/4$, then the perceived difference is $1/4 + N$. Hence, Player 1 cooperates with probability $P(1/4 + N \leq 1/2) = 1/4$ and Player 2 cooperates with probability $P(1/4 + N \leq 3/4) = 1/2$. It turns out that $(\theta_1 = 1, \theta_2 = 1)$, which leads to mutual cooperation with probability 1, is a Nash equilibrium of the meta game. Intuitively, the only way for either player to defect more is to lower their threshold. But then $|\theta_1 - \theta_2|$ will increase, which will cause the opponent to defect more (at a rate of $1/2$). This outweighs the benefit of defecting more oneself.

In Section 4, we prove a folk theorem for diff meta games. Roughly speaking, this result shows that merely observing a diff value is sufficient for enabling all the cooperative outcomes that full mutual transparency enables. Specifically, we show that for every *individually rational* strategy profile $\sigma$ (i.e., every strategy profile that is better for each player than their minimax payoff), there is a diff function such that $\sigma$ is played in an equilibrium of the resulting diff meta game.

Next, we address Obstacle 2 to full mutual transparency – the multiplicity of equilibria. First, note that any given measure of similarity will typically only enable a specific set of equilibria, much smaller than the set of individually rational strategy profiles. For instance, in the above example, all equilibria are symmetric. In general, one would hope that similarity-based cooperation will

|  |  | Player 2 | |
|  |  | Cooperate | Defect |
| Player 1 | Cooperate | $G, G$ | $0, G+1$ |
|  | Defect | $G+1, 0$ | $1, 1$ |

Table 1: The Prisoner's Dilemma, parameterized by some number $G > 1$.

generally result in symmetric outcomes in symmetric games. After all, the new equilibria of the diff game are based on submitting similar policies and if two policies play different strategies against each other, they cannot be similar. In Section 5, we substantiate this intuition. Specifically, we prove, roughly speaking, that in symmetric, additively decomposable games, the Pareto-optimal equilibrium of the meta game is unique and gives both players the same utility, if the measure of difference between the agents satisfies a few intuitive requirements (Section 5). For example, in the Prisoner's Dilemma, the unique Pareto-optimal equilibrium of the meta game must be one in which both players cooperate with the same probability.

Finally we show that diff meta games address Obstacle 3: we demonstrate that in games with higher-dimensional action spaces, we can find cooperative equilibria of diff meta games with ML methods. In Section 8, we show that, if we initialize the two policies randomly and then let each of them learn to be a best response to the other, they generally converge to the Defect-Defect equilibrium. This is expected based on results in similar contexts, such as in the Iterated Prisoner's Dilemma. However, in Section 6, we introduce a novel pretraining method that trains policies to cooperate against copies and defect against randomly generated policies. Our experiments show that policies pretrained in this way find partially cooperative equilibria of the diff game when trained against each other.

We discuss how the present paper relates to prior work in Section 9. We conclude in Section 10 with some ideas for further work.

## 2  BACKGROUND

**Elementary game theory definitions.** We assume familiarity with game theory. For an introduction, see Osborne (2004). A *(two-player, normal-form) game* $\Gamma = (A_1, A_2, \mathbf{u})$ consists of sets of actions or pure strategies $A_1$ and $A_2$ for the two players and a utility function $\mathbf{u} \colon A_1 \times A_2 \to \mathbb{R}^2$. Table 1 gives the Prisoner's Dilemma as a classic example of a game. A mixed strategy for Player $i$ is a distribution over $A_i$. We denote the set of such distributions by $\Delta(A_i)$. We can extend $\mathbf{u}$ to mixed strategies by taking expectations, i.e., $\mathbf{u}(\sigma_1, \sigma_2) \coloneqq \sum_{a_1 \in A_1, a_2 \in A_2} \sigma_1(a_1) \sigma_2(a_2) \mathbf{u}(a_1, a_2)$. For any player $i$, we use $-i$ to denote the other player. We call $\sigma_i$ a *best response* to a strategy $\sigma_{-i} \in \Delta(A_{-i})$, if $\mathrm{supp}(\sigma_i) \subseteq \arg\max_{a_i \in A_i} u_i(a_i, a_{-i})$, where supp denotes the support. A strategy profile $\boldsymbol{\sigma} \in \Delta(A_1) \times \Delta(A_2)$ is a vector of strategies, one for each player. We call a strategy profile $(\sigma_1, \sigma_2)$ a *(strict) Nash equilibrium* if $\sigma_1$ is a (unique) best response to $\sigma_2$ and *vice versa*. As first noted by Nash (1950), each game has at least one Nash equilibrium. We say that a strategy profile $\boldsymbol{\sigma}$ is *individually rational* if each player's payoff is at least its minimax payoff, i.e., if $y_i \geq \max_{\sigma_i \in \Delta(A_i)} \min_{a_{-i} \in A_{-i}} u_i(\sigma_i, a_{-i})$ for $i = 1, 2$. We say that $\boldsymbol{\sigma}$ is *Pareto-optimal* if there exists no $\boldsymbol{\sigma}'$ s.t. $u_i(\boldsymbol{\sigma}') \geq u_i(\boldsymbol{\sigma}')$ for $i = 1, 2$ and $u_i(\boldsymbol{\sigma}') > u_i(\boldsymbol{\sigma}')$ for at least one $i$.

**Symmetric games and additively decomposable games.** We say that a game is *(player) symmetric* if $A_1 = A_2$ and for all $a_1, a_2$ for $i = 1, 2$, we have that $u_i(a_1, a_2) = u_{-i}(a_2, a_1)$. The Prisoner's Dilemma in Table 1 is symmetric. We say that a game *additively decomposes into* $(u_{i,j} \colon A_j \to \mathbb{R})_{i,j \in \{1,2\}}$ if $u_i(a_1, a_2) = u_{i,1}(a_1) + u_{i,2}(a_2)$ for all $i = \{1, 2\}$ and all $a_1 \in A_1, a_2 \in A_2$. Intuitively, this means that each action $a_j$ of Player $j$ generates some amount of utility $u_{i,j}(a_j)$ for Player $i$ *independently* of what Player $-j$ plays. For example, the Prisoner's Dilemma in Table 1 is additively decomposable, where $u_{i,i} \colon \text{Cooperate} \mapsto 0, \text{Defect} \mapsto 1$ and $u_{i,-i} \colon \text{Cooperate} \mapsto G, \text{Defect} \mapsto 0$ for $i = 1, 2$. Intuitively, Cooperate generates $G$ for the opponent and 0 for oneself, while Defect generates 0 for oneself and 1 for the opponent.

**Alternating best response learning.** The orthodox approach to learning in games is to learn to best respond to the opponent, essentially ignoring that the opponent is also a learning agent. In this paper, we specifically consider alternating best response (ABR) learning. In ABR, the players take turns. In each turn, one of the two players updates the parameters $\boldsymbol{\theta}_i$ of her strategy to optimize $u_i(\boldsymbol{\theta}_i, \boldsymbol{\theta}_{-i})$,

i.e., updates her model to be a best response to the opponent's current model (Brown cf. 1951; Heinrich et al. 2021; Zhang et al. 2022). Since learning an exact best response is generally intractable, we will specifically consider the use of gradient ascent in each turn to optimize $u_i(\boldsymbol{\theta}_i, \boldsymbol{\theta}_{-i})$ over $\boldsymbol{\theta}_i$. In continuous games if ABR with *exact* (locally) best response updates converges to $(\boldsymbol{\theta}_1, \boldsymbol{\theta}_2)$, then $(\boldsymbol{\theta}_1, \boldsymbol{\theta}_2)$ is a (local) Nash equilibrium. Note, however, that ABR may fail to converge (e.g., in the face of Rock–Paper–Scissors dynamics). Moreover, if the best response updates of $\theta_i$ are only approximated, ABR may converge to non-equilibria (Mazumdar et al., 2020, Proposition 6).

## 3 DIFF META GAMES

We now formally introduce diff meta games, the novel setup we consider throughout this paper. Given some base game $\Gamma$, we consider a new *meta game* played by two players whom we will call *principals*. Each principal $i$ submits a *policy*. The two players' policies each observe a real-valued measure of how similar they are to each other. Based on this, the policies then output a (potentially mixed) strategy for the base game. Finally, the utility is realized as per the base game. Below we define this new game formally. This model is illustrated in Figure 1.

**Definition 1.** *Let $\Gamma = (A, \mathbf{u})$ be a game. A (diff-based) policy for Player $i$ for $\Gamma$ is a function $\mathbb{R} \to \Delta(A_i)$ mapping the perceived real-valued difference between the policies to a mixed strategy of the game. For $i = 1, 2$ let $\mathscr{A}_i \subseteq \Delta(A_i)^{\mathbb{R}}$ be a set of difference-based policies for Player $i$. Then a policy difference (diff) function for $(\mathscr{A}_1, \mathscr{A}_2)$ is a stochastic function $\mathrm{diff} \colon \mathscr{A}_1 \times \mathscr{A}_2 \rightsquigarrow \mathbb{R}^2$. For any two policies $\pi_1, \pi_2$, we say that $(\pi_1, \pi_2)$ plays the strategy profile $\boldsymbol{\sigma} \in \Delta(A_1) \times \Delta(A_2)$ of $\Gamma$ if $\sigma_i = \mathbb{E}[\pi_i(\mathrm{diff}_i(\pi_1, \pi_2))]$ for $i = 1, 2$. For sets of policies $\mathscr{A}_1, \mathscr{A}_2$ and difference function $\mathrm{diff}$ we then define the diff meta game $(\Gamma, \mathscr{A}_1, \mathscr{A}_2, \mathrm{diff})$ to be the game $(\mathscr{A}_1, \mathscr{A}_2, V)$, where $V(\pi_1, \pi_2) \coloneqq \mathbb{E}[\mathbf{u}((\pi_i(\mathrm{diff}_i(\pi_1, \pi_2)))_{i=1,2})]$ for all $\pi_1 \in \mathscr{A}_1, \pi_2 \in \mathscr{A}_2$.*

Note that Definition 1 does not put any restrictions on diff. For example, the above definition allows $(\mathrm{diff}(\pi_i, \pi_{-i}))_i$ to be a real number whose binary representation uniquely specifies $\pi_{-i}$. This paper is dedicated to situations in which diff specifically represents some intuitive notion of how different the policies are, thus excluding such diff functions. Unfortunately, there are many different ways in which one could formalize this constraint, especially in asymmetric games. In Section 5 we will impose some restrictions along these lines, including symmetry. Our folk theorem (Theorem 3 in Section 4) will similarly impose constraints on diff to avoid diff functions like the above.

The rest of this section will study concrete examples of Definition 1. First, we define a particularly simple type of diff-based policy. Almost all of our theoretical analysis will be based on this class of policies.

**Definition 2.** *Let $\theta \in \mathbb{R} \cup \{-\infty, \infty\}$ and $\sigma_i^{\leqslant}, \sigma_i^{>} \in \Delta(A_i)$ be strategies for Player $i$ for $i = 1, 2$. Then we define $(\sigma_i^{\leqslant}, \theta, \sigma_i^{>})$ to be the policy $\pi$ s.t. $\pi(d) = \sigma_i^{\leqslant}$ if $d \leq \theta$ and $\pi(d) = \sigma_i^{>}$ otherwise. We call policies of this form threshold policies. Let $\bar{\mathscr{A}}_i$ denote the set of such threshold policies.*

Throughout the rest of this section, we analyze the Prisoner's Dilemma as a specific example. We limit attention to threshold agents of the form $(C, \theta, D)$, i.e., policies that cooperate against similar opponents (diff below threshold $\theta$) and defect against dissimilar opponents. This is because such policies can be used to form cooperative equilibria, while policies that always cooperate $((C, 1, C))$ or policies that are more cooperative against *less* similar opponent policies (e.g., $(D, 1, C)$) cannot be used to form cooperative equilibria in the PD with a natural diff function. Policies of the form $(C, \theta, D)$ are uniquely specified by a single real number $\theta$. A natural measure of the similarity between two policies $\theta_1, \theta_2$ is then the absolute difference $|\theta_1 - \theta_2|$. We allow diff to be noisy, however. We summarize this in the following.

**Example 1.** *Let $\Gamma$ be the Prisoner's Dilemma as per Table 1. Then consider the $(\Gamma, \hat{A}_1, \hat{A}_2, \mathrm{diff})$ meta game where $\hat{A}_i = \{(C, \theta_i, D) \mid \theta_i \in \mathbb{R}\}$ and $\mathrm{diff}_i((C, \theta_1, D), (C, \theta_2, D))) = |\theta_1 - \theta_2| + N_i$ for $i = 1, 2$ where $N_i$ is some real-valued random variable.*

The only open parameters of Example 1 are $G$ (the parameter used in our definition of the Prisoner's Dilemma) and the noise distribution. Nevertheless, Example 1 is a rich setting that allows for non-trivial results. We leave a detailed analysis for Appendix B and only give two specific results about equilibria here.

**Proposition 1.** *Consider Example 1 with $N_i \sim$ Uniform$([0,\varepsilon])$ i.i.d. for some $\varepsilon \geq 0$ and with $G \geq 2$. Then $((C,\theta_1,D),(C,\theta_2,D))$ is a Nash equilibrium if and only if $\theta_1,\theta_2 \leq 0$ or $0 < \theta_1 = \theta_2 \leq \varepsilon$. In case of the latter, the equilibrium is strict if $G > 2$.*

Another natural distribution to use for $N_i$ is the normal distribution. The following result shows for $G = 2$ what the Nash equilibria of the diff meta game are.

**Proposition 2.** *Consider Example 1 with $G = 2$. Assume $N_i$ is i.i.d. for $i = 1,2$ according some unimodal distribution with mode $v$ with positive measure on every interval. Then $((C,\theta_1,D),(C,\theta_2,D))$ is a Nash equilibrium if and only if $\theta_1 = \theta_2 \leq v$.*

## 4 A FOLK THEOREM FOR DIFF META GAMES

What are the Nash equilibria of a diff meta game on $\Gamma$? Of course, the answer depends on what diff function we use. A first answer is that Nash equilibria of $\Gamma$ carry over to the diff meta game regardless of what diff function is used (assuming that at least all constant policies are available); see Proposition 15 in Appendix C.1. Any other equilibria of the diff meta game hinge on the use of the right diff function. In fact, if diff is constant and thus uninformative, the Nash equilibria of the diff meta game are exactly the Nash equilibria of $\Gamma$; see Proposition 16 in Appendix C.1. The more interesting question is for what strategy profiles $\boldsymbol{\sigma}$ *there exists* some diff function s.t. $\boldsymbol{\sigma}$ is played in an equilibrium of the resulting diff meta game. The following result answers this question.

**Theorem 3** (folk theorem for diff meta games)**.** *Let $\Gamma$ be a game and $\boldsymbol{\sigma}$ be a strategy profile for $\Gamma$. Let $\mathscr{A}_i \supseteq \bar{\mathscr{A}}_i$ for $i = 1,2$. Then the following two statements are equivalent:*

1. *There is a* diff *function such that there is a Nash equilibrium $(\pi_1,\pi_2)$ of the diff meta game $(\Gamma, \text{diff}, \mathscr{A}_1, \mathscr{A}_2)$ s.t. $(\pi_1,\pi_2)$ play $\boldsymbol{\sigma}$.*
2. *The strategy profile $\boldsymbol{\sigma}$ is individually rational (i.e., better than everyone's minimax payoff).*

*The result continues to hold true if we restrict attention to deterministic* diff *functions with* $\text{diff}_1 = \text{diff}_2$ *and* $\text{diff}_i(\pi_1,\pi_2) \in \{0,1\}$ *for* $i = 1,2$*.*

We leave the full proof to Appendix C.2, but give a short sketch of the construction for 2⇒1 here. For any $\boldsymbol{\sigma}$, we construct the desired equilibrium from policies $\pi_i^* = (\sigma_i, 1/2, \hat{\sigma}_i)$ for $i = 1,2$, where $\hat{\sigma}_i$ is Player $i$'s minimax strategy against Player $-i$. We then take any diff function s.t. $\text{diff}(\pi_i^*, \pi_{-i}) = (0,0)$ if $\pi_{-i} = \pi_{-i}^*$ and $\text{diff}(\pi_i^*, \pi_{-i}) = (1,1)$ otherwise.

## 5 A UNIQUENESS THEOREM

Theorem 3 allows for highly asymmetric similarity-based cooperation. For example, in the PD with, say, $G = 2$, Theorem 3 shows that with the right diff function, the strategy profile $(C, 2/3 * C + 1/3 * D)$ is played in an equilibrium of the diff meta game of the PD. This seems odd, as one would expect similarity-based cooperation to result in playing symmetric strategy profiles. Note that, for example, all equilibria of Propositions 1 and 2 are symmetric. In this section, we show that under some restrictions on diff and the base game $\Gamma$, we can recover the symmetry intuition.

We first need a few definitions of properties of diff. Let $\Gamma$ be a symmetric game. We say that diff is *minimized by copies* if for all policies $\pi, \pi'$, all $y$ and $i = 1,2$, $P(\text{diff}_i(\pi,\pi')<y) \leq P(\text{diff}_i(\pi,\pi)<y)$. For example, the diff function in Example 1 is minimized by copies. The diff functions in the proof of Theorem 3 are not in general minimized by copies when the given base game is symmetric. For example, to achieve $(C, 2/3 * C + 1/3 * D)$ in equilibrium, the proof of Theorem 3 (as sketched above) uses the policies $\pi_1^* = (C, 1/2, D)$ and $\pi_2^* = (2/3 * C + 1/3 * D, 1/2, D)$ and a diff function with $\text{diff}(\pi_1^*, \pi_2^*) = (0,0)$ but $\text{diff}(\pi_1^*, \pi_1^*) = (1,1)$. If the base game is symmetric, we call diff *symmetric* if for all $\pi_1, \pi_2$, $\text{diff}(\pi_1,\pi_2)$ is distributed the same as $\text{diff}(\pi_2,\pi_1)$ and $(\text{diff}_1(\pi_1,\pi_2), \text{diff}_2(\pi_1,\pi_2))$ is distributed the same as $(\text{diff}_2(\pi_1,\pi_2), \text{diff}_1(\pi_1,\pi_2))$.

Finally, we need a more complicated but nonetheless intuitive property of diff functions. In this paper, we generally imagine that *low* values of diff are informative about the other player's policy. In contrast, we will her assume that *high* values of diff are uninformative. That is, for any $\sigma_i$ and $\pi_{-i}$, we will assume that there is a policy $\pi_i$ that plays $\sigma_i$ against $\pi_{-i}$ and triggers the above-threshold policy of $\pi_{-i}$ with the highest-possible probability. Appendix D.1.1 shows why this assumption is necessary. Formally, let $\pi_{-i} = (\sigma_{-i}^{\leq}, \theta_{-i}, \sigma_{-i}^{>})$ be any threshold policy. Let $p$ be the highest number such that there is $\pi_i$ s.t. in $(\pi_i, \pi_{-i})$, Player $-i$ plays arbitrarily close to $(1-p)\sigma_{-i}^{\leq} + p\sigma_{-i}^{>}$ and

$\sigma^{\max}_{\pi_{-i}} = (1-p)\sigma^{\leq}_{-i} + p\sigma^{>}_{-i}$. Intuitively, $\sigma^{\max}_{\pi_{-i}}$ is the strategy played by $\pi_{-i}$ against the most different opponent policies. For the examples of Section 3 we have $p = 1$ and thus simply $\sigma^{\max}_{\pi_{-i}} = \sigma^{>}_{-i}$. But if diff is bounded (as in the proof of Theorem 3), then we might even have $p = 0$ or anything in between.

**Definition 3.** *We call* diff$: \bar{\mathscr{A}}_1 \times \bar{\mathscr{A}}_2 \rightsquigarrow \mathbb{R}^2$ *high value uninformative if for each threshold policy* $\pi_{-i}$, $\sigma_i$ *and* $\varepsilon > 0$ *there is a threshold policy* $\pi_i$ *such that in* $(\pi_i, \pi_{-i})$, *a strategy profile within* $\varepsilon$ *of* $(\sigma_i, \sigma^{\max}_{\pi_{-i}})$ *is played.*

We are now ready to state a uniqueness result for the Nash equilibria of diff meta games.

**Theorem 4.** *Let* $\Gamma$ *be a player-symmetric, additively decomposable game. Let* diff *be symmetric, high-value uninformative, and minimized by copies. Then if* $(\pi_1, \pi_2)$ *is a Nash equilibrium that is not Pareto-dominated by another Nash equilibrium, we have that* $V_1(\pi_1, \pi_2) = V_2(\pi_1, \pi_2)$. *Hence, if there exists a Pareto-optimal Nash equilibrium, its payoffs are unique, Pareto-dominant among Nash equilibria and equal across the two players.*

We prove Theorem 4 in Appendix D.3. Roughly, we prove that under the given assumptions, equilibrium policies are more beneficial to the opponent when observing a diff value below the threshold than if they observe a diff value above the threshold. Second, we show (using the first fact) that if in a given strategy profile Principal $i$ receives a lower utility than Principal $-i$, then Principal $i$ can increase her utility by submitting a copy of Principal $-i$'s policy. Appendix D.1 shows why the assumptions (additive decomposability of the game and and high-value uninformativeness and symmetry of diff) are necessary.

## 6 A NOVEL PRETRAINING METHOD FOR SIMILARITY-BASED COOPERATION

We now describe a simple machine learning method that we use to find cooperative equilibria in more complex games. To use this method, we consider neural net policies $\pi_{\boldsymbol{\theta}}$ parameterized by a real vector $\boldsymbol{\theta}$.

We call this procedure *Cooperate against Copies and Defect against Random (CCDR)* pretraining based on its intended effect in Prisoner's Dilemma-like games. First, for any given diff game, let $V^d \colon (\mathbb{R}^m)^{(\mathbb{R}^{n+1})} \times (\mathbb{R}^m)^{(\mathbb{R}^{n+1})} \to \mathbb{R}^2$ be the utility of a version of the game in which diff is non-noisy. Then we will pretrain each model $\pi_{\theta_i}$ to maximize $V^d(\pi_{\theta_i}, \pi_{\theta_i}) + V^d(\pi_{\theta_i}, \pi'_{\theta_{-i}})$ for randomly sampled $\theta'_{-i}$. That is, each player $i$ pretrains their policy $\pi_{\theta_i}$ to do well in both of the following scenarios: principal $-i$ copies principal $i$'s model; and principal $-i$ generates a random model.

CCDR pretraining is motivated by two considerations. First, in games like the HDPD, it can be shown that there exist cooperative equilibria between policies that cooperate at a diff value of 0 and defect as the perceived diff value increases. We give a toy model of this in Appendix E. CCDR puts in place the rudimentary structure of these equilibria. Note, however, that CCDR does not directly optimize for the model's ability to form a cooperative equilibrium. Second, CCDR can be thought of as a form of curriculum training. Before trying to play diff games against other (different but similar) learned agents, we might first train a policy to solve two (conceptually and technically) easier related problems.

## 7 A HIGH-DIMENSIONAL ONE-SHOT PRISONER'S DILEMMA

To study similarity-based cooperation in an ML context, we need a more complex version of the Prisoner's Dilemma. The complex Prisoner's Dilemma-like games studied by the multi-agent learning community generally offer other mechanisms that establish cooperative equilibria (e.g., playing a game repeatedly). For our experiments, however, we specifically need SBC to be the only mechanism to establish cooperation.

We therefore introduce a new game, the High-Dimensional (one-shot) Prisoner's Dilemma (HDPD). The goal is to give a variant of the one-shot Prisoner's Dilemma that is conceptually simple but introduces scalable complexity that makes finding, for example, exact best responses in the diff meta game intractable. In addition to $G$, the HDPD is parameterized by two functions $f_C, f_D \colon \mathbb{R}^n \to \mathbb{R}^m$ representing the two actions Cooperate and Defect, respectively, as well as a probability measure $\mu$ over $\mathbb{R}^n$. Each player's action is also a function $f_i \colon \mathbb{R}^n \to \mathbb{R}^m$. This is illustrated in Figure 2 for

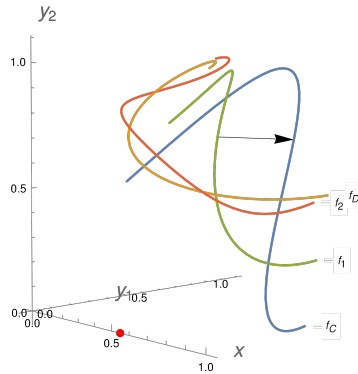

Figure 2: The figure describes how utilities are calculated in the HDPD. Here the functions $f_1$ and $f_2$ are the actions chosen by the players. First we sample a point $x$ (red). We then calculate the distance of $f_i(x)$ to $f_C(x)$ and $f_D$ to determine both players' losses.

the case of $n = 1$ and $m = 2$. For any pair of actions $f_1, f_2$, payoffs are then determined as follows. First, we sample some $\mathbf{x}$ according to $\mu$ from $\mathbb{R}^n$. In Figure 2, this is represented by a red dot on the x axis. Then to determine how much Player 1 cooperates, we consider the distance $d(f_1(\mathbf{x}, f_C(\mathbf{x}))$ to determine, roughly speaking, how much Player 1 cooperates. Here, $d$ denotes the Euclidean distance. The larger the distance the less cooperative is $f_1$. This distance is visualized in Figure 2 by the arrow from $(\mathbf{x}, f_1(\mathbf{x}))$ to $(\mathbf{x}, f_C(\mathbf{x}))$. We analogously determine how much the players defect. Formally, we define $u_i(f_1, f_2) = -\mathbb{E}_{\mathbf{x} \sim \mu} [d(f_i(\mathbf{x}), f_D(\mathbf{x})) + G d(f_{-i}(\mathbf{x}), f_C(\mathbf{x}))] / \mathbb{E}_{\mathbf{x} \sim \mu} [d(f_C(\mathbf{x}), f_D(\mathbf{x}))]$. Thus, the action $f_i = f_D$ corresponds to defecting and the action $f_C$ corresponds to cooperating, e.g., $\mathbf{u}(f_C, f_C) = (-1, -1)$ and $\mathbf{u}(f_D, f_D) = (-G, -G)$. The unique equilibrium of this game is $(f_D, f_D)$. In our experiments, we specifically used $G = 5$. If we further let $\mu$ be uniform over $[0, 1]$, the utilities for Figure 2 come out at about $-5.501$ for Player 1 and $-2.599$ for Player 2.

We consider a diff meta game on the HDPD. Formally, a diff-based policy for the HDPD is a function $\mathbb{R} \to (\mathbb{R}^m)^{(\mathbb{R}^n)}$. For notational convenience, we will instead write policies as functions $\mathbb{R}^{n+1} \to \mathbb{R}^m$. We then define our diff function by $\mathrm{diff}_i(\pi_1, \pi_2) = \mathbb{E}_{(y, \mathbf{x}) \sim \nu} [d(\pi_1(y, \mathbf{x}), \pi_2(y, \mathbf{x}))] + N_i$, where $\nu$ is some probability distribution over $\mathbb{R}^{n+1}$ and $N_i$ is some real-valued noise.

## 8 EXPERIMENTS

Our results so far demonstrate the theoretical viability of similarity-based cooperation, but leave open questions regarding its practicality. In this section, we address one of these questions: In complex environments, where cooperating and defecting are by themselves complex operations, can we find the cooperative equilibria for a given diff function with machine learning methods?

**Experimental setup.** We trained on the environment from section 7. We selected a fixed set of hyperparameters based on prior exploratory experiments and the theoretical considerations in Appendix E. We then randomly initialized $\theta_1$ and $\theta_2$ and CCDR-pretrained them. Finally, we trained the $\theta_1$ and $\theta_2$ against each other using ABR. We repeated the experiment with 28 different seeds. As control, we also ran the experiment *without* CCDR pretraining (on 26 seeds). We also ran experiments with Learning with Opponent-Learning Awareness (LOLA) (Foerster et al., 2018), which we report in Appendix G.

**Results.** First, we observe that in the runs without CCDR pretraining, the players generally converge to mutual defection during alternating best response learning. In particular, in all 26 runs, at least one player's utility was below $-5$. Only two runs had a utility above $-5$ for one of the players ($-4.997$ and $-4.554$). The average utility across the 26 runs and across the two players was $-5.257$ with a standard deviation of $0.1978$. Anecdotally, these results are robust – ABR without pretraining practically never finds cooperative equilibria in the HDPD.

Second, we observe that in all 28 runs, CCDR pretraining qualitatively yields the desired policy models, i.e., a policy that cooperates at low values of diff and gradually comes closer to defecting at high values of diff. Figure 3a shows a representative example.

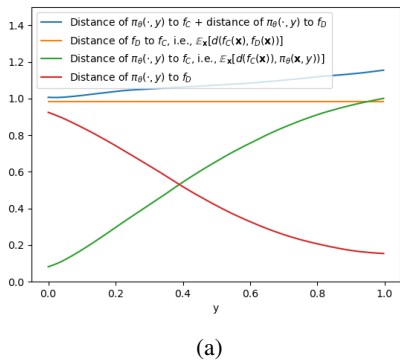
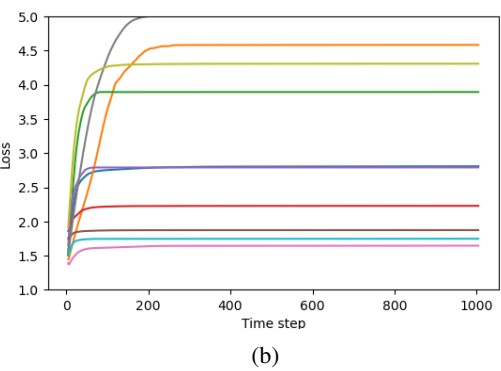

(a)                                        (b)

Figure 3: (a) The behavior of a CCDR pretrained policy. For each perceived diff to the opponent $y$, the graph shows the expected distance of the learned policy's choice to to $f_C$ and to $f_D$. (b) Losses of Player 1 in 10 runs through the ABR phase.

Our main positive experimental result is that after CCDR pretraining, the models converged in alternating best response learning to a partially cooperative equilibrium in 26 out of 28 runs. Thus, the cooperative equilibria postulated in general by Theorem 3 and in simplified examples by Propositions 1 and 2 (as well as Proposition 24), do indeed exist and can be found with simple methods. The minimum utility of either player across the 26 successful runs was -4.854. The average utility across all runs and the two players was about -2.77 and thus a little closer to $u(f_C, f_C) = -1$ than to $u(f_D, f_D) = -5$. The standard deviation was about 1.19. Figure 3b shows the losses (i.e., the negated utilities) across ABR learning. Generally, the policies also converge to receiving approximately the same utility (cf. Section 5). The average of the absolute differences in utility between the two players at the end of the 28 runs is about 0.04 with a standard deviation of 0.05. We see that in line with Theorem 4, we tend to learn egalitarian equilibria in this symmetric, additively decomposable setting. After alternating best response learning, the models generally have a similar structure as the model in Figure 3a, though often they cooperate only a little at low diff values. Based on prior exploratory experiments, CCDR's success is moderately robust.

**Discussion.** Without pretraining, ABR learning unsurprisingly converges to mutual defection. This is due to a bootstrapping problem. Submitting a policy of the form "cooperate with similar policies, defect against different policies" is a unique best response if the opponent submits a model of this form as well. If the opponent model $\pi_{-i}$ is not of this form, then any policy $\pi_i$ that defects, i.e., that satisfies $\pi_i(\text{diff}(\pi_i, \pi_{-i})) = f_D$, is a best response. Because $f_C$ is complex, learning a model that cooperates at all is unlikely. (Even if $f_C$ was simple, the appropriate use of the perceived diff value would still be specific and thus unlikely to be found by chance.) Similar failures to find the more complicated cooperative equilibria by default have also been observed in the iterated PD (Sandholm & Crites 1996; Foerster et al. 2018; Letcher et al. 2019) and in the open-source PD (Hutter, 2020) (cf. Section 9.1). Opponent shaping methods have been used successfully to learn to cooperate both in the iterated Prisoner's Dilemma (Foerster et al. 2018; Letcher et al. 2019) and the open-source Prisoner's Dilemma (Hutter, 2020). Our experiments in Appendix G show that LOLA can also learn SBC, but unfortunately not as robustly as CCDR pretraining.

CCDR pretraining reliably finds models that cooperate with each other and that continue to partially cooperate with each other throughout ABR training. This shows that when given some guidance, ABR can find similarity-based cooperative equilibria. We conclude from our experiments that SBC is a suitable means of establishing cooperation between modern ML agents.

That said, CCDR also has some limitations that we hope can be addressed in future work. For one, in many games optimal play against randomly generated opponents is unreasonable when facing a rational opponent. Second, our experiments show that while the two policies almost fully cooperate after CCDR pretraining, they quickly partially unlearn to cooperate in the ABR phase. We would prefer a method that preserves closer to full cooperation throughout ABR-style training. Third, while CCDR seems to often work, it can certainly fail in games in which SBC is possible. For instance, CCDR may sometimes result in insufficiently steep incentive curves. We suspect that to

make progress on the latter issues we need training procedures that more explicitly reason about incentives *à la* opponent shaping (cf. our experiments with LOLA Appendix G).

## 9 RELATED WORK

### 9.1 PROGRAM EQUILIBRIUM

We already discussed in Section 1 the literature on program meta games in which players submit computer programs as policies and the programs fully observe each other's code (McAfee 1984; Howard 1988; Rubinstein 1998, Section 10.4; Tennenholtz 2004). Interestingly, some constructions for equilibria in program meta games are similarity based. For example, the earliest cooperative program equilibrium for the Prisoner's Dilemma, described in all four of the above-cited papers, is the program "Cooperate if the opponent's program is equal to this program; else Defect". Other approaches to program equilibrium cannot be interpreted as similarity based, however (see, e.g., Barasz et al., 2014; Critch, 2019; Oesterheld, 2019b). To our knowledge, the only published work on ML in program equilibrium is due to Hutter (2020). It assumes the programs to have the structure proposed by Oesterheld (2019b) on simple normal-form games, thus leaving only a few parameters open. Similar to our experiments, Hutter shows that best response learning fails to converge to the cooperative equilibria. In Hutter's experiments, the opponent shaping methods LOLA (Foerster et al., 2018) and SOS (Letcher et al., 2019) converge to mutual cooperation.

### 9.2 DECISION THEORY AND NEWCOMB'S PROBLEM

Brams (1975) and Lewis (1979) have pointed out that the Prisoner's Dilemma against a similar opponent closely resembles *Newcomb's problem*, a problem first introduced to the decision-theoretical literature by Nozick (1969). Most of the literature on Newcomb's problem is about the normative, philosophical question of whether one should cooperate or defect when playing a Prisoner's Dilemma against an exact copy. Our work is inspired by the idea that in some circumstances one should cooperate with similar opponents. However, this literature only informally discusses the question of whether to also cooperate with agents other than exact copies (Hofstadter e.g., 1983; Drescher 2006, Ch. 7; Ahmed 2014, Sect. 4.6.3). We address this question formally.

One idea behind the present project, as well as the program game literature, is to analyze a decision situation from the perspective of (actual or hypothetical) *principals* who design policies. The principals find themselves in an ordinary strategic situation. This is how our analysis avoids the philosophical issues arising in the *agent's* perspective. Similar changes in perspective have been discussed in the literature on Newcomb's problem (Gauthier e.g., 1989; Oesterheld & Conitzer 2022).

### 9.3 LEARNING IN NEWCOMB-LIKE DECISION PROBLEMS

There is some existing work on learning in Newcomb-like environments that therefore also applies to the Prisoner's Dilemma against a copy. Whether cooperation against a copy is learned generally depends on the learning scheme. Bell et al. (2021) show that $Q$-learning with a softmax policy learns to defect. Regret minimization also learns to defect. Other learning schemes do converge to cooperating against exact copies (Albert & Heiner, 2001; Mayer et al., 2016; Oesterheld, 2019a; Oesterheld et al., 2021). All schemes in prior work differ from the present setup, however, and to our knowledge none offer a model of cooperation between similar but non-equal agents.

## 10 CONCLUSION AND FUTURE WORK

We make a strong case for the promise of similarity-based cooperation as a means of improving outcomes from interactions between ML agents. At the same time, there are many avenues for future work. On the theoretical side, we would be especially interested in generalizations of Theorem 4, that is, theorems that tell us what outcomes we should expect in diff meta games. Is it true more generally that under reasonable assumptions about the diff function, we can expect similarity-based cooperation to result in fairly specific, symmetric, Pareto-optimal outcomes? We are also interested in further experimental investigations of SBC. We hope that future work can improve on our results in the HDPD in terms of robustness and degree of cooperation. Besides that, we think a natural next step is to study settings in which the agents observe their similarity to one another in a more realistic fashion. For example, we conjecture that similarity-based cooperation can occur when the agents can determine that their policies were generated by similar learning procedures.

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
