# OpenReview forum: "Similarity-Based Cooperation"
_ICLR.cc/2023/Conference — Submitted to ICLR 2023_

### Official Review · Reviewer_sbJz · 2022-10-23

**Confidence:** 4
**Clarity, Quality, Novelty And Reproducibility:** Paper is well written. Clarity could …
**Correctness:** 4
**Technical Novelty And Significance:** 3
**Empirical Novelty And Significance:** 3
**Recommendation:** 5

**Strength And Weaknesses:**

Mathematically rigorous results are obtained and the experiments are done to back up the claims. The paper presents contributions for enhancing similarity based cooperative learning between ML agents and clearly explains how their approach presents solutions to difficulties with existing methods.
Paper compares full transparency setting vs their approach and explains how the proposed method addresses problems of existing methods in great detail. The authors could consider explaining the results more in a more accessible way to broader audience and elaborate connections to representation learning.
The paper is dense and needs more clarifications.

Minor: You have used a distance measure as the similarity measurement. Does the direction matter? i.e. is diff(θ1,θ2) = diff(θ2,θ1) in general or should the measurement change according to the perceiver?

**Summary Of The Paper:**

This paper presents a new way to allow cooperation between ML agents using an approach based on a similarity measurement. Authors introduce a program called diff meta games, a novel variant of program meta games, where agents observe the similarity/dissimilarity between the policies using a single number, thus the similarity/ dissimilarity between each other, allowing the agents to learn to cooperate/defect. Authors theoretically and experimentally show the validity of their method of similarity-based cooperation. They prove that this approach allows the same cooperative outcomes as in the full mutual transparency setting. Moreover, they experimentally show that cooperative learning through this approach can be achieved with simple ML methods.

**Summary Of The Review:**

The authors study an interesting problem with simulating motivation. Interesting theoretical and experimental results are obtained.

Explaining the results more in a more accessible way to broader audience and elaborating connections to representation learning could improve the quality of the paper.

---

> ### Author Response · Authors · 2022-11-11
> **Response to review**
>
> > Minor: You have used a distance measure as the similarity measurement. Does the direction matter? i.e. is diff(θ1,θ2) = diff(θ2,θ1) in general or should the measurement change according to the perceiver?
>
> Good question! The equality diff(θ1,θ2) = diff(θ2,θ1) does not necessarily hold and this is somewhat important for the paper. (Perhaps more precisely: diff(θ1,θ2) need not be equally distributed to diff(θ2,θ1).) For one, if the game has different action spaces for the two players, then diff(θ2,θ1) will be undefined, because θ2 and θ1 have the wrong types. Second, the folk theorem (Theorem 3) requires the use of diff functions for symmetric games that are asymmetric in this way, i.e., diff functions such that diff(θ1,θ2) ≠ diff(θ2,θ1). All examples in the main text consider symmetric games with diff(θ1,θ2) = diff(θ2,θ1). Also, in Section 5 we explicitly assume diff(θ1,θ2) = diff(θ2,θ1). We clarified this a bit in Section 3 after Definition 1 and in Section 5, where we introduce symmetry.
>
> >connections to representation learning
>
> We would be interested to hear if the reviewer has more specific thoughts, as we are unsure which potential connections the reviewer has in mind. It may be that agents could be encouraged to learn an internal representation of their similarity to other agents. In that case, this could be used as a similarity measure, instead of an externally provided similarity measure. This would be an interesting angle for future work. We will add a brief discussion of this point to the paper.

---

### Official Review · Reviewer_diwd · 2022-10-25

**Confidence:** 3
**Correctness:** 3
**Technical Novelty And Significance:** 3
**Empirical Novelty And Significance:** 2
**Recommendation:** 5

**Clarity, Quality, Novelty And Reproducibility:**

Could the authors provide a discussion about how their work is related to works in the following line of research?

__Assortative matchmaking__

Some papers (e.g. Wang et al 2018, https://arxiv.org/pdf/1811.05931.pdf) shows that the mechanism of matching individuals who have similar behaviour can encourage cooperation in social dilemmas (and even sequential social dilemmas). Wang et al (2018) gave a matchmaking system in which before an episode, agents measure their similarity (with respect to the rewards) to each other, and similar agents will be matched together. Pro-social behaviour can be observed under this mechanism. The authors also discussed the theoretical ground of this method. The authors can also see many citations to previous methods on matchmaking in this paper.

The reviewer noticed the some differences between the proposed method and matchmaking. However, the reviewer also believes that the two ideas are similar in spirit.

__Homophily__

Homophily is a very well-studied approach in the field of cooperation emergence. It has shown that if agents are encouraged to be similar to each other, cooperation is more likely to emerge. One insight is that homophily can solve the issue of second order social dilemma. The reviewer just lists some related areas where the concept of homophily has been investigated. For example:
- [1] Di Stefano, Alessandro, et al. "Quantifying the role of homophily in human cooperation using multiplex evolutionary game theory." PloS one 10.10 (2015): e0140646.
- [2] Aksoy, Ozan. "Effects of heterogeneity and homophily on cooperation." Social Psychology Quarterly 78.4 (2015): 324-344.


The authors may also want to discuss the prior findings in evolutionary game theory. For example,

-Christian Hilbe, Štěpán Šimsa, Krishnendu Chatterjee, Martin A Nowak, “Evolution of cooperation in stochastic games.”

**Strength And Weaknesses:**

The paper provides a clear discussion of the motivation and details of the proposed method.

The reviewer has a major concern about the relationship between the proposed method and the large body of literature on homophily and assortative matchmaking (see the section below).

**Summary Of The Paper:**

This paper studies promoting cooperation in games. The authors provide a method where similarity information can achieve similar cooperation levels compared to previous method that requires agent's policies to be transparent.

**Summary Of The Review:**

The reviewer generally agree this paper makes some contribution to the cooperation learning in social dilemmas, but has concerns about its position in the literature.

---

> ### Author Response · Authors · 2022-11-11
> **Response to review**
>
> Thanks for sharing these very interesting papers! We were not aware of these before. However, we don’t think any of them call into question the novelty of our work.
>
> Re Wang et al 2018: I wasn’t aware of the idea of assortative matchmaking before. It’s quite a clever mechanism for establishing cooperation. However, I don’t think it is very similar to the idea of our paper. Ultimately, assortative matchmaking seems to belong to the tit-for-tat/reputation effects cluster of methods of establishing cooperation. In all of these cases, cooperation can be achieved (in equilibrium) because the game is repeated and at each point the decision to cooperate (or defect) can have an impact on whether future opponents will cooperate with you. This seems fundamentally different from the commitment cluster of methods, to which similarity-based cooperation belongs.
>
> Re Di Stefano et al 2015: As far as I understand, the mechanism of establishing cooperation in this example hinges on the mechanism described by Nowak and May (1992): “Evolutionary games and spatial chaos” https://www.nature.com/articles/359826a0 (as cited by Di Stefano et al. (2015)). This mechanism is different from both SBC and cooperation in repeated games via tit for tat. Note, for example, that it hinges on the spatial component of a many-player setup. It does not apply to two-player games (unless the two-player game is transformed into a many player game by playing on a lattice). Also note that none of these papers study Nash equilibria. Instead they study the dynamics resulting from very specific learning dynamics.
>
> Re Aksoy (2015): This paper studies homophily as a behavior in humans, specifically the tendency of humans to cooperate with more similar individuals. Indeed, this sociological phenomenon in humans could be related to similarity-based cooperation. Therefore, we will add a discussion of this to our paper. That said, there are many possible explanations for humans’ tendency to cooperate with more similar individuals. In fact, the explanation given by Aksoy (2015) seems very different from the analysis of our paper. Also, we are not aware of any game-theoretic analysis of the mechanism we discuss in our paper, nor of any machine learning experiments using this mechanism.
>
> Hilbe et al. also exclusively discusses tit-for-tat-style cooperation in repeated games. We will clarify that our paper is not about tit-for-tat-style cooperation.

---

### Official Review · Reviewer_ATts · 2022-10-25

**Confidence:** 2
**Correctness:** 3
**Technical Novelty And Significance:** 2
**Empirical Novelty And Significance:** 2
**Recommendation:** 6

**Clarity, Quality, Novelty And Reproducibility:**

The paper is dense and, while technically precise, lacks clarity. Much detail is left for the appendix and the reader's effort. Such is typical of game theoretic papers, but, this paper would have a greater impact if the details were better explained.

The results roughly and empirically support the theoretical analysis, but a definitive "proof positive" result in the empirical section is missing. The results are also difficult to parse.

The work appears novel.

Code is provided in the supplementary materials.



**Strength And Weaknesses:**

Strengths:
+ The paper clearly states its contributions. These contributions include the introduction a specific game called "diff meta games," and theoretical analysis regarding these games. This analysis leads to the claimed contribution that cooperative equilibria can be found with ML methods.
+ The paper gives an appropriate if dense set of preliminaries in Section 2.
+ The intuition/motivation added in the second to last paragraph of Section 6 is helpful.
+ Section 8's discussion section is helpful.

Weaknesses:
-Figure 1 is, unfortunately, not beneficial. It appears to setup a typical prisoner's dilemma (Figure 1a). Figure 1b is not explained in the text. These figures should be removed or better explained (or illustrated differently).
-Table 1 does not show the only formulation of a Prisoner's Dilemma -- "G+1" could be any number k > G.
- The definitions (e.g., Definition 1 and 2) are not explained from an intuitive perspective or with a set of working examples until Example 1. This reviewer recommends each definition add intuition to unpack what is going on rather than waiting for Example 1. Even example 1, however, is more of an illustrative set of cases rather than a grounded example with actual numbers (e.g., "N_i is some real-valued random variable").
-"Appendix C.2, but give" should not have a comma.
-The pretraining procedure in Section 6 is only briefly described. More details could/should be given.
-The value of Figure 2 is unclear.
-Figure 3 is poorly formatted and difficult to parse.
-More discussion of limitations would be helpful. For example, how large can games be (rows and columns of the normal form game formulation) before the method breaks down?
-The result that a single number being shared can achieve cooperation is perhaps not as impressive if one considers that the number could be a real-valued number with an infinite number of digits. Can the nature of this number be restricted?
-What happens if agent(s) are byzantine?

**Summary Of The Paper:**

Overall, this paper provides a unique perspective for the ICLR community -- developing theoretical insights into how cooperative agents might function. The paper focuses on develop the theory to support the sufficiency of agents sharing minimal information to achieve cooperation in a Prisoner's Dilemma setup. Theoretical and empirical results are presented.

**Summary Of The Review:**

This paper provides theoretical and empirical results investigating Prisoner Dilemma-like games and whether agents might be cooperative under minimal information sharing. The theoretical analysis is a strong positive for the paper (plus the empirical results); however, the paper is difficult to parse. Without a more approachable presentation, this paper might have limited impact. The reviewer acknowledges limited expertise in game theory (while being an expert in ML).

---

> ### Author Response · Authors · 2022-11-11
> **Response to review**
>
> We thank the reviewer for the especially detailed suggestions for how to make the paper clearer! We will work to clarify the paper as suggested.
>
> >how large can games be (rows and columns of the normal form game formulation) before the method breaks down?
>
> I assume “the method” refers to CCDR. We already test CCDR on a game with uncountably many rows and columns (the High-Dimensional Prisoner’s Dilemma (HDPD)). I assume the question is about one of the following two issues.
>
> While the HDPD has infinitely large action spaces, one might view it as a version of the PD, which is quite simple. For example, the HDPD still has a dominant action (f_D). So a natural question to ask is what happens if we consider a large normal-form game, say, without dominant actions and with multiple Nash equilibria. We could even consider a high-dimensional version of that game (in which one needs to compute some function to compute each of the available actions). Whether CCDR pretraining still gives desirable results depends a little on the game. In general, we think the CC (“cooperate against copies”) part of CCDR pretraining makes sense in general, though one may run into issues with local maxima of the CC part of the loss (e.g., one may learn to play (Hare, Hare) with a copy in some versions of Stag Hunt). The DR part would need to be adapted because in general the best response to the random opponent opponent distribution is not a very reasonable strategy in the game. So one might instead train against some other learning model to learn to play a Nash equilibrium strategy against dissimilar opponents. We will add a brief discussion of the points of this paragraph to the “Discussion” part of Section 8.
>
> In the submitted version of the paper, we have not provided any information on the computational costs of our experiments. So the reader is currently unable to judge how efficiently our method works on our toy games and whether there might be room to scale it to more complex games. This is simply an oversight. We will make sure to add information on this to the paper. Here’s a short overview: In the experiments, the CCDR training took a few minutes per run on a single CPU. (Training with LOLA as done in our experiments also only takes minutes per run on a single CPU.) The main constraint on making our experiments higher-dimensional/more complex is that the ABR training phase takes a lot of time (a few hours per run). This has two reasons. First, we sacrifice a lot of efficiency to ensure that ABR is very stable (making small steps, taking only successful steps). Second, we aim to train to convergence to make sure that we learn Nash equilibria and avoid spurious positive results.
>
> >The result that a single number being shared can achieve cooperation is perhaps not as impressive if one considers that the number could be a real-valued number with an infinite number of digits. Can the nature of this number be restricted?
>
> Yes! We discuss this a bit in various places, e.g., in the text right after Definition 1. The last sentence of Theorem 3 also answers this question, but we now realize that this is a bit cryptic, so we clarified it. Of course, all of our specific examples also have only “natural” measures of similarity.
>
> >What happens if agent(s) are byzantine?
>
> Good question! If the agents are byzantine, it may be harder to obtain the needed signal of similarity in the real world. (Our paper focuses on the case of heterogeneous signals of similarity and therefore has little to say about the extent to which this is the case.) However, one strength of similarity-based cooperation (relative to full-mutual-transparency-based program equilibrium) is that the complexity of the agents is not necessarily an obstacle to similarity-based cooperation. For example, if an agent learns that its opponent is equal to itself, then this is always sufficient for achieving cooperation, even when the agents do not understand their own or the opponent’s model. I would argue that our experiments also show the effectiveness of similarity-based cooperation in the case of byzantine agents. Presumably, the models in our experiments (thousands of parameters) are difficult to understand by just inspecting them. Nevertheless, similarity-based cooperation can be achieved.

---

> > ### Comment · Reviewer_ATts · 2022-11-11
> > **Response**
> >
> > I appreciate the reviewer's response. Can the authors proceed with updating their paper based upon the proposal to "We will add a brief discussion of the points of this paragraph to the “Discussion” part of Section 8" so that I can review the changes?
> >
> > It would be helpful to add information -- even to the appendix -- to remedy the limitation that "The reader is currently unable to judge how efficiently our method works on our toy games and whether there might be room to scale it to more complex games."

---

> > > ### Author Response · Authors · 2022-11-16
> > > **Response to reviewer**
> > >
> > > >I appreciate the reviewer's response. Can the authors proceed with updating their paper based upon the proposal to "We will add a brief discussion of the points of this paragraph to the “Discussion” part of Section 8" so that I can review the changes?
> > >
> > > Done!
> > >
> > > >It would be helpful to add information -- even to the appendix -- to remedy the limitation that "The reader is currently unable to judge how efficiently our method works on our toy games and whether there might be room to scale it to more complex games."
> > >
> > > We added notes on how long the experiments took to compute to Appendix G.4 and F.6.

---

> ### Author Response · Authors · 2022-11-11
> **Response to review (continued)**
>
> >The results roughly and empirically support the theoretical analysis, but a definitive "proof positive" result in the empirical section is missing.
>
> It would be helpful for us if the reviewer could elaborate on how our empirical results fall short of this. We do believe that the empirical results show quite definitely that similarity-based cooperation can be leveraged with simple ML methods. Of course, the results have various limitations, as we also point out in the paper. However, almost all papers on multi-agent learning in general-sum games have similar limitations (due to the hardness of some of the problems). So it would be helpful for us to know which limitation of the results specifically the reviewer thinks this paper ought to address.

---

> > ### Comment · Reviewer_ATts · 2022-11-11
> > **Clarification**
> >
> > This reviewer appreciates the authors' note about the perceived ambiguity of the statement. To clarify, it would be helpful if the paper's discussion section could note where in the empirical results the reader can observe the consequences of the various theorems in the paper. Narratively, the theroetical and empirical analyses appear disjoint.

---

> > > ### Author Response · Authors · 2022-11-16
> > > **Response to reviewer**
> > >
> > > This is a very helpful suggestion! We clarified these connections in the “Results” part of Section 8. (The connection to Propositions 1 and 2 and Theorem 3 is that all of these results indicate the existence of cooperative SBC equilibria. Our experimental results show that these equilibria indeed exist and moreover can be found in practice. The connection to Theorem 4 is that Theorem 4 suggests a bias of SBC toward egalitarian, symmetric solutions, which our experimental results confirm.)

---

> > > > ### Comment · Reviewer_ATts · 2022-11-20
> > > > **Feedback**
> > > >
> > > > Based upon the authors' response and changes, I am increasing my score.

---

### Official Review · Reviewer_cK7h · 2022-10-27

**Confidence:** 3
**Clarity, Quality, Novelty And Reproducibility:** Original work.
**Correctness:** 3
**Technical Novelty And Significance:** 2
**Empirical Novelty And Significance:** 3
**Recommendation:** 5

**Strength And Weaknesses:**

Strengths:
1.	A similarity-based cooperation mechanism is proposed.
2.	Efficient theorems have been proposed to analyze the Nash equilibrium property of diff meta games.
Weaknesses:
1.	Full transparency are rare in reality, however, similarity is also not an explicit property which can be obtained by observation, how to evaluate the similarity between players in fact?
2.	It is a basic question, why players prefer to cooperate with similar players? Which lacks sufficient explanations.
3.	A similarity-based cooperation mechanism can promote cooperation, what will happen if the other player do not adopt the similarity-based cooperation mechanism, for example, the other player uses the TFT strategy. On the other word, I am interested in the robustness of the similarity-based cooperation mechanism.
4.	The words in Fig. 3(b) is too small to read, which should be improved.

**Summary Of The Paper:**

One way to enable cooperative outcomes is to make the agents mutually transparent to each other, however, full transparency is often unrealistic. In this paper, authors introduce a more realistic setting in which agents only observe how similar they are to each other. It is proved that this allows for the same set of cooperative outcomes as the full transparency setting.

**Summary Of The Review:**

One way to enable cooperative outcomes is to make the agents mutually transparent to each other, however, full transparency is often unrealistic. In this paper, authors introduce a more realistic setting in which agents only observe how similar they are to each other. It is proved that this allows for the same set of cooperative outcomes as the full transparency setting.
Strengths:
1.	A similarity-based cooperation mechanism is proposed.
2.	Efficient theorems have been proposed to analyze the Nash equilibrium property of diff meta games.
Weaknesses:
1.	Full transparency are rare in reality, however, similarity is also not an explicit property which can be obtained by observation, how to evaluate the similarity between players in fact?
2.	It is a basic question, why players prefer to cooperate with similar players? Which lacks sufficient explanations.
3.	A similarity-based cooperation mechanism can promote cooperation, what will happen if the other player do not adopt the similarity-based cooperation mechanism, for example, the other player uses the TFT strategy. On the other word, I am interested in the robustness of the similarity-based cooperation mechanism.
4.	The words in Fig. 3(b) is too small to read, which should be improved.

---

> ### Author Response · Authors · 2022-11-11
> **Response to review**
>
> Re 1: Good question! Certainly in the real world, similarity is not directly observed as a single real number. However, we believe that in many realistic deployment settings, AI systems will have information available to assess how similar they are to each other. For example, it may be known that two AI systems were generated by the same or very similar training processes. In the present work we focus on the case of real-valued differences because it allows for simpler analysis. (Perhaps this is clearest on the theoretical side, but it also simplifies the analysis of our experiments. For instance, the real-valued signal allows us to visualize an agent’s behavior as per Figure 3a.) As we note in the conclusion of our paper, we think that developing methods to establish similarity-based cooperation based on more realistic signals of similarity (say, signals about the similarity of the training processes) is one of the most important directions for future work on this topic.
>
> Re 2: We wouldn’t say that players prefer to cooperate with similar players. Instead, mutual cooperation is preferred by both players over mutual defection. And cooperating with similar players and defecting against dissimilar players is the only way to achieve this in an equilibrium. (For instance, cooperating via tit for tat is not possible in the problems under consideration, because they are single-shot. Moreover, “cooperate with dissimilar players” wouldn’t be an equilibrium strategy as it is not an optimal response to a player that always defects.) This is why we focus on policies that cooperate with similar players. We added some explanation of this between Definition 2 and Example 1 in Section 3.
>
> Re 3: The settings we study are all single-shot and therefore don’t allow for tit-for-tat. (We discuss in Section 7 why we made this choice. If our experiments allowed for multiple mechanisms of establishing cooperation, the results would be harder to interpret.) If one player doesn’t use the similarity-based cooperation mechanism, then there’s not much the other player can do, so she will defect.
>
> If we did consider a setting in which tit-for-tat-style cooperation is also possible, then using similarity-based cooperation wouldn’t interfere with also using tit for tat. That is, one can (and probably should in most cases) build agents that use both mechanisms. Of course, if one player trains their agent only to use similarity-based cooperation and the other trains their agent only to use tit for tat, then the two agents would typically not cooperate with each other.

---

### Author Response · Authors · 2022-11-11
**General comment**

We thank the reviewers for their efforts in evaluating and helping us improve our paper! We will post replies to the individual reviews. One theme throughout multiple reviews is that the paper is hard to understand. The reviews already contain helpful suggestions on how to make the paper more accessible, but any further comments on this would be greatly appreciated. We have already addressed some specific comments (see the individual responses) and will work over the paper further for the camera ready copy.

---

> ### Author Response · Authors · 2022-11-16
> **Revision**
>
> We have uploaded a revised version of our paper which implements some of the changes suggested by the reviewers.

---

### Decision · Program_Chairs · 2023-01-20

**Decision:**

Reject

**Justification For Why Not Higher Score:**

- Weakness of empirical justification
- Sensitivity of the proposed method to the choice of diff function, along with weak evidence that an appropriate diff function can be created for problems beyond the high-dimensional prisoner's dilemma
- Lack of clarity and accessibility to an ICLR audience

**Justification For Why Not Lower Score:**

N/A

**Metareview: Summary, Strengths And Weaknesses:**

This paper considers one-shot games with transparency. In this previously-studied setting, there is a two-player game (e.g., Prisoner's Dilemma) to be played by two algorithms. Before playing the game, the algorithms submit their code to a trusted third party. Algorithm A's (mixed) policy can depend on Algorithm B's code, and vice versa. Past work shows that this setting allows many more equilibria (and better equilibria that support cooperation) than without transparency. Unfortunately, it is hard to make this setting practical, e.g., because of the high-dimensional nature of code as a description and the difficulty of processing this input.

This paper proposes a *potentially* more practical formulation in which Algorithm A only receives a signal diff(Algo A, Algo B) describing the similarity between its algorithm and that of Algorithm B. Algorithm B receives the same signal. Much of the paper is focused on threshold policies, where one policy is followed when the similarity score is above a threshold, and another when it is below.

The paper then shows that this setting supports cooperation. In particular, any individually rational policy (i.e., one that is better than minimax optimal for all players, and that players might wish to follow given the opportunity to commit) is an equilibrium under an appropriate method for computing differences (Theorem 3).

The paper then shows that, under a set of natural properties on the method for computing differences, Pareto-optimal Nash equilibria have unique payoffs that are symmetric across players (Theorem 4).

The paper then proposes a method for finding cooperative equilibria, i.e., policies such that the best response results in an equilibrium whose payoffs are good for both players. This consists of optimization of parameters in a neural net (which generates the policy's actions)  that, intuitively, seeks to have a large reward when playing both against copies of itself and against randomly generated policies. This novel method is called CCDR. Two policies generated in this way are then supplied as input to alternating best response to find an equilibrium.

The paper then demonstrates this method on a high-dimensional prisoner's dilemma. It finds that using its novel CCDR method allows the players to find cooperative equilibria often (though not all of the time) while doing alternating best response on random policies does not find cooperative equilibria.

Strengths
- Novel approach. While cooperation supported by full-transparency has been studied, and there is existing work on homophily, this paper seems to be the first to consider the power of similarity signals in this way.
- The idea of passing only a similarity rather than the full source code is appealing --- as the paper shows, it offers the same power as the full transparency setting, but is much simpler.

Weaknesses
- The paper is not written in a way that make it widely accessible to an ICLR audience. It is mathematically precise but important definitions and statements are sometimes not accompanied by clear intuitive explanations. Reviewer ATts gives a good summary and examples of these weaknesses.
- The choice of diff function seems critical to creating good cooperative equilibria that can be practically computed. Yet, the construction of such diff functions is not explored well. An existence result for diff functions with cooperative equilibria is given in Theorem 3, but it does not seem practical.  A specific diff function is given for the high-dimensional prisoner's dilemma but this is not general, nor is this choice explored.
- The empirical demonstration of the method is limited to the high-dimensional prisoner's dilemma. While interesting, this game does not seem practically important.
- The paper does not articulate its contributions in the context of the literature on homophily and assortative matchmaking. One of the reviewers felt quite strongly that this was a significant weakness of the paper. The AC feels that this literature is related and should be discussed, but that the approach taken in the current paper is different, as articulated by the authors in the rebuttal.

**Summary Of Ac-Reviewer Meeting:**

Points raised in the meeting:
- Reviewers felt that the lack of clarity in the paper made it quite difficult for them to judge its quality. Some had less experience with mathematical game theoretic arguments.
- We had a discussion of homophily and the novelty of the paper relative to this literature. We spent some time unpacking this. After discussion, we agreed that the paper's model & theoretical results were novel. It was simply that some of the motivation and insights overlapped with the homophily literature.
- We had a discussion of the lack of practical justification and evidence that the ideas in the paper are useful.

In making my final decision, I discounted to some extent the concerns about clarity because some of this came from the reviewers having less experience with game theoretic papers. At the same time, the paper could have been much clearer. I took much more seriously my own reading of the paper, the weakness of its empirical evaluation, and the sensitivity to the choice of the diff function.